# Prognostic Value of PD-L1, PD-1 and CD8A in Canine Diffuse Large B-Cell Lymphoma Detected by RNAscope

**DOI:** 10.3390/vetsci8070120

**Published:** 2021-06-29

**Authors:** Luca Aresu, Laura Marconato, Valeria Martini, Antonella Fanelli, Luca Licenziato, Greta Foiani, Erica Melchiotti, Arturo Nicoletti, Marta Vascellari

**Affiliations:** 1Department of Veterinary Sciences, University of Turin, 10095 Grugliasco, Italy; antonella.fanelli@unito.it (A.F.); luca.licenziato@unito.it (L.L.); arturo.nicoletti@unito.it (A.N.); 2Department of Medical Veterinary Science, University of Bologna, 40064 Ozzano dell’Emilia, Italy; laura.marconato@unibo.it; 3Department of Veterinary Medicine, University of Milan, 26900 Lodi, Italy; valeria.martini@unimi.it; 4Istituto Zooprofilattico Sperimentale delle Venezie (IZSVe), 35020 Legnaro, Italy; gfoiani@izsvenezie.it (G.F.); emelchiotti@izsvenezie.it (E.M.); mvascellari@izsvenezie.it (M.V.)

**Keywords:** dog, diffuse large B-cell lymphoma, PD-1, PD-L1, RNAscope

## Abstract

Immune checkpoints are a set of molecules dysregulated in several human and canine cancers and aberrations of the PD-1/PD-L1 axis are often correlated with a worse prognosis. To gain an insight into the role of immune checkpoints in canine diffuse large B-cell lymphoma (cDLBCL), we investigated PD-L1, PD-1 and CD8A expression by RNAscope. Results were correlated with several clinico-pathological features, including treatment, Ki67 index and outcome. A total of 33 dogs treated with chemotherapy (*n* = 12) or chemoimmunotherapy with APAVAC (*n* = 21) were included. PD-L1 signal was diffusely distributed among neoplastic cells, whereas PD-1 and CD8A were localized in tumor infiltrating lymphocytes. However, PD-1 mRNA was also retrieved in tumor cells. An association between PD-L1 and PD-1 scores was identified and a higher risk of relapse and lymphoma-related death was found in dogs treated with chemotherapy alone and dogs with higher PD-L1 and PD-1 scores. The correlation between PD-L1 and PD-1 is in line with the mechanism of immune checkpoints in cancers, where neoplastic cells overexpress PD-L1 that, in turn, binds PD-1 receptors in activated TIL. We also found that Ki67 index was significantly increased in dogs with the highest PD-L1 and PD-1 scores, indirectly suggesting a role in promoting tumor proliferation. Finally, even if the biological consequence of PD-1+ tumor cells is unknown, our findings suggest that PD-1 intrinsic expression in cDLBCL might contribute to tumor growth escaping adaptive immunity.

## 1. Introduction

Diffuse large B-cell lymphoma (DLBCL) is the most frequent histotype in dogs, accounting for 50–60% of all non-Hodgkin lymphomas [1]. Despite its relative morphological and phenotypical homogeneity, DLBCL encompasses multiple clinical entities in dogs, and only a small number of animals is definitively cured by treatment [1,2,3,4,5].

Recently, compelling evidence has established the importance of NF-κB signaling in canine DLBCL (cDLBCL), partially modeling the human activated B-cell-like DLBCL (ABC-DLBCL) [6,7]. In one study using RNA and methylation sequencing, two subgroups of cDLBCL with distinct clinical behavior and molecular features were identified. The signatures characterizing the two subgroups did not overlap with the human germinal center (GC) and ABC-DLBCL subtypes, but were largely defined by host response and microenvironment mechanisms, including immune checkpoints. Among genes involved in T-cell regulation, programmed death 1 receptor (PD-1), PD Ligand-1 (PD-L1), and CD8A were up-regulated in the subgroup with a poor prognosis [8].

PD-1 is a well-known immune checkpoint molecule expressed by chronically stimulated CD4 and CD8 cells. PD-L1 is one of the two ligands for PD-1, and the PD-1/PD-L1 axis is critical in terminating immune responses by exhausting self-reactive T cells, thereby protecting against autoimmunity [9]. While playing a key role in physiological immune homeostasis, the PD-1/PD-L1 axis also putatively serves as a means through which cancer cells evade the immune system. In human neoplasia, PD-L1 is reported to be expressed by tumor cells, whereas PD-1 is generally expressed by tumor-infiltrating lymphocytes (TIL), and their interaction negatively regulates the adaptive antitumor immune response [10]. The development of anti-PD-1/PD-L1 monoclonal antibodies has recently become a hot topic in cancer immunotherapy, thereby dramatically changing the therapeutic approaches for many cancers at an advanced stage [11]. Favorable long-term outcomes with these therapies have been reported in lung and breast cancer; however, despite the promising results, only 20–40% of patients respond and have durable disease remission [12]. Therefore, assays to identify patients who are most likely to benefit from these agents are critical [13].

In human DLBCL, PD-1 is mainly expressed by immune cells and promotes T-cell exhaustion and immuno-tolerance. Conversely, PD-L1 is expressed by tumor cells and is associated with a short survival and in vitro proliferation of malignant B-cells. So far, few clinical trials using anti-PD-1 and anti-PD-L1 monoclonal antibodies showed promising results and moderate adverse effects [14].

In dogs, a wide range of solid and hematologic malignancies exploit the PD-1/PD-L1 axis, and the development of monoclonal antibodies targeting immune checkpoints is in the pipeline [15,16]. A pilot clinical study evaluating the clinical efficacy of a chimeric anti-PD-L1 monoclonal antibody in canine oral malignant melanoma and undifferentiated sarcoma demonstrated uncertain results [17]. Recently, Igase M. and colleagues developed two bioactive anti-PD-1 monoclonal antibodies providing evidence of safety, tolerability and antitumor efficacy in dogs with different histotypes [18]. Correspondingly, commercially available assays to detect the presence of PD-L1 and PD-1 proteins in tumors have gathered interest, but cross-specific antibodies are scarcely available in dogs, and thus alternative methods are needed [19].

RNAscope is a fully automated, in situ hybridization (ISH) assay for the detection of RNA in a variety of samples, including formalin-fixed paraffin-embedded (FFPE) tissues. This technology derives from a unique probe design and amplification system that is able to potentially amplify target-specific signals without any background interference. Successful hybridization of the probe on the targeted nucleic acid initiates a cascade that leads to deposition of a chromogen, characterized by small dots, which are quantitatively measurable [20]. 

Here, we applied RNAscope to evaluate PD-1, PD-L1 and CD8A expression in cDLBCLs. Furthermore, we correlated the results with clinico-pathological data, including treatment, Ki67 index, TIL, immune gene expression signatures, and outcome.

## 2. Materials and Methods

### 2.1. Animals

cDLBCL samples were retrieved from the archive of the Canine Lymphoma Biobank [21]. For each dog, the following clinico-pathological features were available: signalment, clinical stage, substage, flow cytometric (FC) assessment of lymph node (LN), peripheral blood (PB) and bone marrow (BM) infiltration, serum LDH level, whether dogs had been pre-treated with steroids, type of treatment (standardized chemotherapy or chemoimmunotherapy with APAVAC), time to progression (TTP), lymphoma-specific survival (LSS), and cause of death. By immunohistochemistry (IHC), Ki67 index was calculated in neoplastic lymph nodes as the number of positive cells per 1000 randomly selected cells excluding necrotic areas and data was expressed as percentage [22]. To calculate TIL, CD3 positive cells were counted in a 2.37 mm^2^ area, and results were categorized as follows: 0 (0–5 TIL), 1 (6–15 TIL), 2 (16–25 TIL), 3 (26–40 TIL) and 4 (>40 TIL) [22].

### 2.2. Transcriptomic Immune Landscape of DLBCL

Since RNA-seq data were available for all tumors, we calculated the immune landscape as previously described [8]. Briefly, we retrieved immune gene sets for immune cells, TIL, proinflammatory molecules, cytokines and cytokine receptors, regulatory T-cells and immune checkpoints, and performed single sample GSEA to derive the enrichment score of each immune-related term [23]. By applying unsupervised consensus clustering analysis, we separated cDLBCL into two subgroups (high immunity = “hot” and low immunity = “cold”). Moreover, the expression levels of PD-1, PD-L1 and CD8A measured as log counts per million (logCPM) reads were retrieved.

### 2.3. PD-1, PD-L1 and CD8A RNAscope Assay

RNAscope assay was performed according to the manufacturer’s instructions on the Ventana Discovery Ultra autostainer (Ventana Medical System, Roche) using three RNAscope^®^ 2.5 vs. specific Probes (Advanced Cell Diagnostics Inc, Santa Monica, CA, USA): Cl-CD274 (PD-L1; Cat. No. 488469), targeting the region 283–1237 of the canine PD-L1 mRNA (Accession No: NM_001291972.1); CI-CD8A (Cat. No. 459549) targeting the region 41–900 of the canine CD8A mRNA (Accession No: NM_001002935.2); CI-PDCD1 (Cat. No. 488499), targeting the region 75–756 of the canine PD-1 mRNA (Accession No: NM_001002935.2). Briefly, 4 µm thick FFPE serial sections were deparaffinized and pre-treated with heat and protease before hybridization. Each probe was incubated at 42 °C for 2 h. The final deposit was detected as a red, punctate precipitate using the mRNA RED Detection Kit (Roche cod. 07099037001). For each sample, 2 further sections were stained using probes for Fc-PPIB [peptidylprolyl isomerase B (cyclophilin B)] and dapB (Bacillus subtilis dihydrodipicolinate reductase gene). PPIB was used as an endogenous control to assess RNA integrity, while the bacterial gene dapB served as a negative control to assess background staining.

### 2.4. RNAscope Semi-Quantitative Analysis

The semi-quantitative Advanced Cell Diagnostics scoring system for the RNAscope^®^ assay was used [24]. Briefly, the average number of dots per cell was measured and a score of 0 was given when no staining or less than 1 dot for every 10 cells was found, a score of 1 when number of dots was comprised between 1 to 3 per cell, a score of 2 when number of dots was comprised between 4 to 9 per cell, a score of 3 when number of dots was comprised between 10 to 15 per cell and/or a percentage <10% of dots were in clusters, and a score of 4 when number of dots was higher than 15 per cell and/or a percentage >10% of dots were in clusters (Figure 1). The score was evaluated in a 2.37 mm^2^ area, excluding necrotic areas. RNAscope scores were obtained independently by three pathologists (LA, AN, LL) who were blind to the clinical-pathological data. Furthermore, for PD-1, the tumor proportion score (TPS) and the immune cell density score (IDS) was calculated. TPS was obtained as the percentage of PD-1+ tumor cells relative to all cells present in a 2.37 mm^2^ area. The IDS was obtained as the percentage of PD-1+ immune cells relative to all the cells present in a 2.37 mm^2^ area. Discordant results were reviewed, and the final scores were obtained by consensus.

### 2.5. Statistical Analysis

Continuous variables were tested for normal distribution by means of a Shapiro–Wilk test. Thereafter, a Kruskal–Wallis test was used to assess possible differences in a Ki67 index among different RNAscope scores and TIL groups. The correlation between CD8A RNA-seq data and the percentage of CD8+ cells in LN detected via FC was investigated via a Spearman rank correlation test. Multinomial logistic regression was used to assess possible influence of pre-treatment with corticosteroids on RNAscope scores. Contingency tables were prepared and the Pearson chi-square test was performed to assess possible associations between results of the different RNAscope assays and between each RNAscope assay and immune signature. The following variables were tested for their influence on TTP and LSS by means of univariate and multivariate Cox proportional hazard model: breed (pure or mixed), sex (female, spayed female, male, neutered male), age (<7 years, ≥7 years), weight (<10 kg, ≥10 kg), stage (III, IV, V), substage (a, b), PB infiltration (%), BM infiltration (%), presence of BM infiltration (yes, no), FC CD8+ cells (%), LDH activity (normal, increased), pretreatment with steroids (yes, no), treatment (chemotherapy, chemoimmunotherapy with APAVAC), immune signature by RNA-seq (hot, cold), PD-L1 logCPM, PD-1 logCPM, CD8A logCPM, PD-L1 RNAscope score, PD-1 RNAscope score, CD8A RNAscope score, Ki67 index, TIL group. Variables with a *p*-value ≤ 0.300 were included in the multivariate analysis. For categorical variables, Kaplan–Meier curves were drawn and compared by means of log-rank test. TTP was calculated from the start of treatment to disease progression [25]. Dogs lost to follow-up or dead for lymphoma-unrelated causes before disease progression were censored for TTP analysis. LSS was measured as the interval between the start of treatment and death from lymphoma [25]. Dogs dead for lymphoma-unrelated causes were censored for survival analysis.

## 3. Results

### 3.1. Animals 

Thirty-three dogs with DLBCL were included in the analysis (Figure 2M,N). Signalment, clinical presentation, clinico-pathological variables, staging work-up for each case are listed in Appendix A. There were 7 (21.2%) mixed-breed dogs and 26 (78.8%) pure-breed dogs, among which German shepherd (*n* = 4, 15.4%) and Rottweiler (*n* = 3, 11.5%) were the most commonly represented. There were 17 (51.5%) females (11 spayed) and 16 males (3 neutered). Median age was 7 years (range, 3–15 years) and median weight was 26 kg (range, 21–50 kg). Overall, 9 (27.3%) dogs had received prednisone before being referred. At the time of diagnosis, 2 (6.1%) dogs had stage III disease, 6 (18.2%) dogs had stage IV disease, and 25 (75.7%) dogs had stage V disease. Overall, 23 (69.7%) dogs were asymptomatic (substage a), whereas 10 (30.3%) dogs showed clinical signs (substage b). Mean PB infiltration was 5.4 ± 11.2% (median, 1.1%; range, 0.1–55.5%). Mean BM infiltration was 6.7 ± 12.5% (median, 1.4%; range 0.5–50.0%). Fourteen (42.4%) dogs had an increased LDH serum activity. Regarding treatment, 12 (36.4%) dogs received the same CHOP-based chemotherapeutic protocol, whereas 21 (63.6%) dogs were treated with chemoimmunotherapy with APAVAC. The mean percentage of CD8+ cells in LN was 3.7 ± 4.3% (median, 2.3%; range, 0.0–16.9%), whereas mean Ki67 index was 41.6 ± 24.8% (median, 34.0%; range, 9.0–95.0%). TIL groups were distributed as follows: 9 (27.3%) dogs in group 0, 5 (15.2%) in group 1, 11 (33.3%) in group 2, 7 (21.2%) in group 3 and 1 (3.0%) in group 4. 

Regarding RNA-seq enrichment analysis, 19 (57.6%) dogs showed a hot immune signature and 14 (42.4%) a cold one. The mean transcript amount for PD-L1 was 4.1 ± 1.3 logCPM (median, 4.1 logCPM; range, 2.3–7.9 logCPM); 1.1 ± 2.2 logCPM (median, 1.3 logCPM; range, −2.5–6.5 logCPM) for PD-1 and 2.2 ± 2.2 logCPM (median, 1.9 logCPM; range, −2.3–6.5 logCPM) for CD8A.

FC infiltration of CD8+ cells was not associated to RNA-seq results (*p* = 0.281 and *p* = 0.159, respectively). Ki67 index did not vary among TIL groups (*p* = 0.258 and *p* = 0.770, respectively). 

### 3.2. RNAscope Semi-Quantitative Evaluation 

Overall, 33 cDLBCLs were positive for PD-L1, 26 for PD-1 and 31 for CD8A. All the samples incubated with the PPIB probe showed good mRNA integrity and were negative with the DapB control probe. The RNAscope signal was detected as multiple relatively small red dots with nuclear and perinuclear cellular distribution. RNAscope scores were not influenced by pre-treatment with corticosteroids (*p* = 0.729 for PD-L1, *p* = 0.089 for PD-1 and *p* = 0.515 for CD8A). PD-L1 signal was diffusely distributed among neoplastic cells (Figure 2A,D,G,L), whereas PD-1 (Figure 2B,E,H) and CD8A (Figure 2C,F,I) signals showed multifocal distribution retrieved in small lymphocytes ascribable to TIL. Interestingly, in all 26 cDLBCLs expressing PD-1, an mRNA signal was also identified in neoplastic cells, and both TPS and IDS were calculated. For IDS, the mean percentage was 12.9% (range, 4–30%). For TPS the mean percentage was 4.1% (range, 1–7%).

The RNAscope score distribution among samples is reported in Figure 3. The statistical analysis showed a significant association between PD-L1 and PD-1 expression (*p* = 0.002), while no association was observed between PD-L1 and CD8A (*p* = 0.426), as well as between PD-1 and CD8A (*p* = 0.805). Moreover, both PD-L1 and PD-1 expression resulted in being significantly associated with Ki67 index (*p* < 0.001 for PD-L1 and *p* = 0.006 for PD-1), as well as with the RNA-seq immune signatures (*p* = 0.005 for PD-L1 and *p* = 0.033 for PD-1). CD8A expression was significantly associated with TIL groups (*p* < 0.001) but not with FC CD8+ infiltration (*p* = 0.159). Finally, no correlation was retrieved for TPS and IDS.

### 3.3. Survival Analysis 

Thirty-one out of 33 (93.9%) dogs experienced disease progression within the end of the study, whereas 2 (6.1%) died for lymphoma-unrelated causes after 76 and 1403 days from diagnosis, respectively. Median TTP was 171 days (range, 1–1403 days). 

At the end of the study, all dogs had died. Cause of death was attributable to lymphoma in 30 (90.9%) dogs and was unrelated in 3 (9.1%) animals. Median LSS was 237 days (range, 22–1403 days). Median TTP and LSS and results of survival analysis are shown in Appendix A.

A shorter TTP was observed in neutered males, dogs treated with chemotherapy alone, dogs with a hot immune signature and those with higher PD-L1 (Figure 4A) and PD-1 scores. Substage, treatment, PD-1 logCPM and PD-L1 score remained significant at multivariate analysis (Appendix A).

A shorter LSS was observed in dogs treated with chemotherapy alone (Figure 4B), dogs with a hot immune signature (Figure 4C) and a higher PD-L1 (Figure 4D) and PD-1 score. Only PD-L1 score and treatment were significant at multivariate analysis (Appendix A).

## 4. Discussion

Understanding molecular and genetic mechanisms that control the host response to tumors has led to the discovery of immune checkpoints [26]. In several human hematological malignancies, the PD-1/PD-L1 pathway is exploited by tumor cells to evade antitumor immune responses, and ultimately progress and disseminate. Thus, the development and application of inhibitors that block PD-1/PD-L1 interaction result in durable responses and prolonged survival in patients with lymphoma [27]. However, only a subset of patients benefit from treatment and the selection of the best candidates remains an unanswered clinically relevant question. Currently, the most promising response predictors include PD-L1 and PD-1 mRNA or protein analysis, microsatellite instability and tumor mutational burden [28]. 

Recent studies suggest that the PD-1/PD-L1 axis also plays a pivotal role in a wide range of canine malignancies, including oral melanoma, osteosarcoma, hemangiosarcoma, mast cell tumor, and mammary carcinoma [29,30,31,32,33,34,35]. In canine B-cell lymphoma, a higher expression of PD-L1 by neoplastic lymphocytes compared to normal B-cells was demonstrated by flow-cytometry, and an increased in vitro drug resistance was associated with PD-1 and PD-L1 protein expression [28]. 

In the present study, we applied an RNAscope ISH assay using three specific canine probes. PD-L1, PD-1 and CD8A mRNA was investigated in FFPE samples of cDLBCL in order to explore prognostic significance and correlations with other clinico-pathological features. As expected, PD-L1 signal was localized only in neoplastic centroblasts and immunoblasts, whereas PD-1 and CD8A were mainly retrieved in small lymphocytes ascribable to tumor-infiltrating T-cells. This result is in line with the well-known mechanism of immune checkpoints in cancers, where neoplastic cells overexpress PD-L1 that, in turn, binds PD-1 receptors in activated TIL. The interaction counteracts the TCR-signaling cascade and impairs T-cell activation. The correlation between PD-L1 and PD-1 scores identified in our study further supports this hypothesis and provides new evidence on the role of the tumor microenvironment in cDLBCLs [36,37,38].

In the future, the expression of both molecules may be also considered as a valid marker of the immune signature activation status of cDLBCLs. In fact, dogs with the highest PD-L1 and PD-1 mRNA signals were characterized by a hot immune signature revealed by transcriptome analysis. It is noteworthy that RNAscope is less expensive compared to RNA-seq and it can be applied on FFPE tissue samples, being a consistent advantage in both diagnostic and research fields. This technology maps the mRNA signals into individual cells and allows the integration of molecular information with histopathological features. 

In our study, we found that Ki67 index was significantly increased among dogs with higher PD-L1 and PD-1 scores. The interaction between PD-L1 and PD-1 is known to promote tumor proliferation, most likely indirectly. Indeed, PI3K/AKT/mTOR pathway is the major key regulator of PD-L1 and, when aberrantly up-regulated, it stimulates tumor cell growth and proliferation. In canine B-cell lymphoma and specifically in cDLBCL, the mechanism might be similar, since PI3K/AKT/mTOR pathway is frequently deregulated [8,39]. However, functional studies are needed to confirm this hypothesis. Unfortunately, the lack of a standardization of protocols and methods for assessing Ki67 expression in canine lymphoma limits the relevance of this result in the routinary practice. Here, to be consistent with previous data, we assessed a Ki67 index similarly to the score system previously applied in canine Burkitt-like lymphoma [22].

Interestingly, PD-1 was also expressed in neoplastic cells (i.e., centroblasts or immunoblasts) as demonstrated by TPS. While the role of PD-1 in leukocytes is well established across species, the biological mechanisms of PD-1 aberrant expression by tumor cells are still unknown in canine tumors. Recent investigations in human medicine have revealed both genetic and epigenetic factors, such as copy number aberrations or methylation aberrancies, contributing to the activation of this checkpoint in cancer cells [40]. In few human cancers, the direct activation of the PD-1 axis by cancer cells determines a more potent inhibitory signal in T-cells. Specifically, it has been demonstrated that PD-1 activation by tumor cells inhibits glucose consumption, cytokine production, proliferation and survival of T-cells [40]. We could not demonstrate this hypothesis in our experiment, but this aspect deserves in-depth studies in the future.

Here, we also investigated CD8A mRNA levels in cDLBCLs. The transcript was absent in neoplastic cells and only localized in areas characterized by small lymphocytes compatible with CD3+ cells. Indirectly, this was confirmed by the correlation between CD8A scores and TIL groups. Conversely CD8A scores were not correlated to the percentage of CD8+ T-cells by FC, but this was expected since fine needle aspiration sampling was randomly performed in the neoplastic lymph nodes, whereas CD8A scores were obtained, examining the same lymph node section used to count TILs.

Clinically, we observed that a higher expression of PD-L1 mRNA by tumor cells was associated with a shorter survival and disease free-interval. The same was true for PD-1, but only for TTP. Thus, PD-L1 and PD-1 expression might help to identify cDLBCL cases with a higher risk of disease progression. When stratifying outcome based on treatment, chemoimmunotherapy with APAVAC significantly improved TTP and LSS, as already documented. In detail, dogs treated with chemotherapy alone, those with a “hot” immune signature and dogs with higher PD-L1 and PD-1 scores, presented both a higher risk of progression and lymphoma-related death.

Interestingly, increased PD-L1 score was associated with a higher risk of progression and lymphoma-related death regardless of treatment. This result is in line with recent evidence in humans, suggesting that upregulation of PD-L1 in tumor cells allows tumors to elude the host’s immune system and increase chemoresistance. The mechanism is not completely clear, but it is noteworthy that up-regulation of PD-L1 might be caused by several pro-oncogenic pathways, such as MAPK, PI3K/Akt as well as transcriptional factors STAT3 and NF-ĸB. Finally, even if not proven here, it might be possible that dogs relapsing after treatment might have an increased expression of both immune checkpoints as previously demonstrated in vitro, thereby reducing the effect of a rescue treatment and negatively impacting survival [28].

Although we did our best to analyze as many dogs as possible, we acknowledge that the population is too small to fully assess the effects of these markers on survival. In the future we are aiming to enroll more patients to assess them also at the protein level, and to evaluate their PD-L1 score at admission, in order to exclude from chemoimmunotherapy with APAVAC dogs with a poor prognosis, which will not benefit from expensive treatments.

In conclusion, this study indicates that PD-L1 and PD-1 mRNA expression can be reliably assessed by RNAscope in FFPE tissue and may be considered as valid prognostic markers in cDLBCL. In both clinical and research fields, RNAscope may represent a good alternative to IHC, whose reliability is affected by the lack of canine-specific antibodies, and standardized procedures. As a matter of fact, discordant results were recently obtained for PD-L1 in canine lymphoma by using different antibodies [28]. Conversely, RNAscope uses a unique probe design strategy targeting mRNA and provides standardized protocols and reagents, thus limiting inter-laboratory variability.

In human medicine, PD-L1 and PD-1 assessed by IHC are commonly used to predict outcome and treatment response to immune checkpoint monoclonal antibodies, but the data are still controversial. Several methods of assessment and definitions of positivity for both molecules have been proposed, but a consensus has not been reached yet. Here, we also found a correlation of PD-L1 and PD-1 scores with treatment response, Ki67 index and immune signatures. All together, these results suggest that the PD-1/PD-L1 pathway contributes to cDLBCL pathogenesis and may be an effective therapeutic target to be considered in the future.

## Figures and Tables

**Figure 1 vetsci-08-00120-f001:**
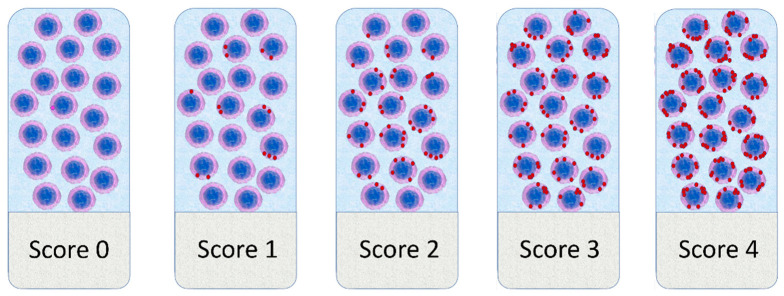
Graphical representation of the scoring system adopted for RNAscope results evaluation.

**Figure 2 vetsci-08-00120-f002:**
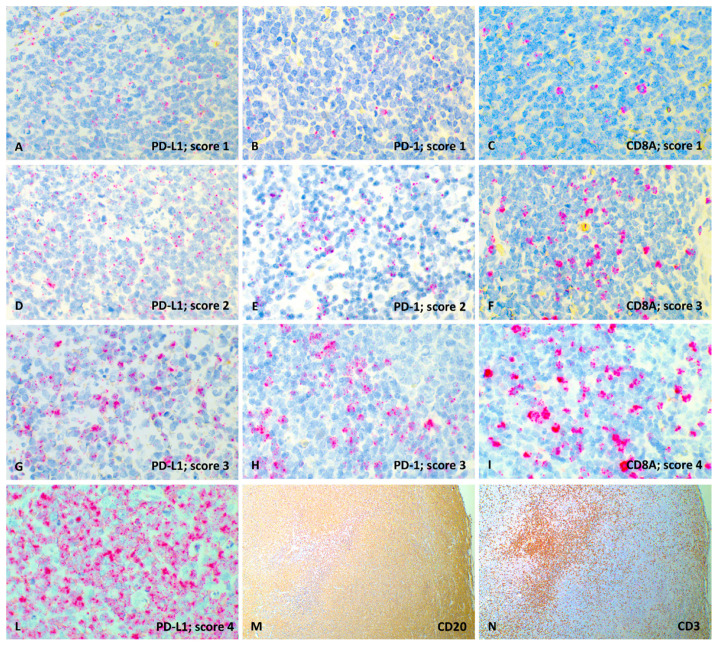
Representative images of PD-L1, PD-1 and CD8A RNAscope semi-quantitative evaluation (60× magnification) and CD20, CD3 immunohistochemical expression (4× magnification), lymph node, dog. (**A**) PD-L1 probe, score 1; (**B**) PD-1 probe, score 1; (**C**) CD8A probe, score 1; (**D**) PD-L1 probe, score 2; (**E**) PD-1 probe, score 2; (**F**) CD8A probe, score 3; (**G**) PD-L1 probe, score 3; (**H**) PD-1 probe, score 3; (**I**) CD8A probe, score 4; (**L**) PD-L1 probe, score 4; (**M**) Diffuse and strong membranous labeling of neoplastic lymphocytes with anti-CD20 antibody (IHC); (**N**) Focal and strong cytoplasmatic labeling of TIL with anti-CD3 antibody (IHC).

**Figure 3 vetsci-08-00120-f003:**
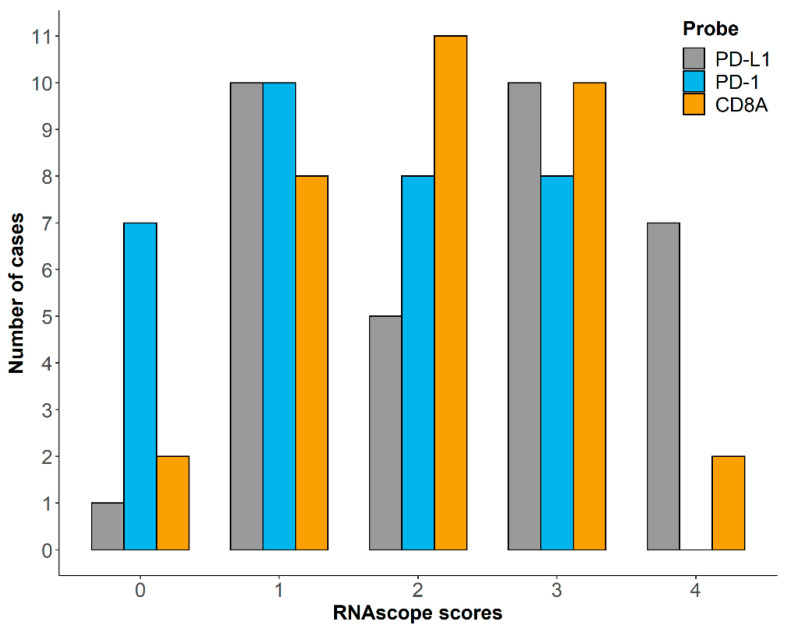
Distribution of RNAscope scores for PD-L1, PD-1 and CD8A among 33 cDLBCL samples.

**Figure 4 vetsci-08-00120-f004:**
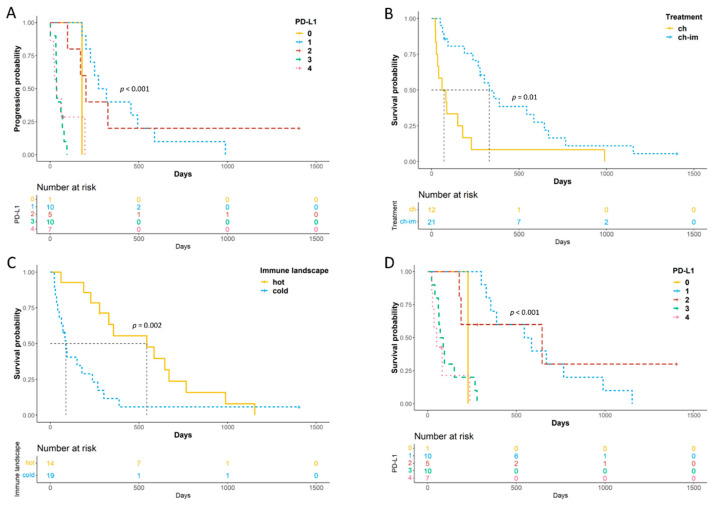
Kaplan–Meier (KM) for TTP and LSS of 33 dogs with cDLBCL. (**A**) KM curve of TTP according to PD-L1 RNAscope scores; (**B**,**C**) KM curves of LSS according to treatment (**B**) and immune signature enrichment status (**C**); (**D**) KM curve of LSS according to PD-L1 RNAscope scores.

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
