# Peer review of "Prognostic Value of PD-L1, PD-1 and CD8A in Canine Diffuse Large B-Cell Lymphoma Detected by RNAscope"

_vetsci, 2021, doi:10.3390/vetsci8070120_

Round 1
Reviewer 1 Report
The authors present in the manuscript very interesting findings regarding canine DLBCL expression of PD1, PDL1 and CD8a , which was evaluated using RNAscope technology. The manuscript is overall well written and the analysis is well performed. There are though some elements that should be added both in the text, as well as statistical analyses.
- in the introduction, please add information regarding clinical trials studying PD1/PDL1 blockade in human DLBCL and how successful (or not) they were; also add information regarding the PD1/PDL1 antibodies developed to date and clinical trials in dogs
- for data analysis- please include in statistical analysis of also subdivided fractions of PD1 expression (tumour cells and TILs separately). Also add a separate graph showing PD1 expression on tumour cells/TILs. This can actually have a potential clinical relevance if one day PD1/PDL1 therapy will be introduced to veterinary oncology, as PD1 can serve as both tumour suppressor as well as an oncogene, depending on cancer type ( https://doi.org/10.1080/2162402X.2017.1408747 ; 10.1016/j.cell.2015.08.052) .The discussion should be expanded to include the potential tumourigenic effects of PD1/PDL1. Additional information, on effect of PDL1 expression on resistance to chemotherapy should also be included and information from this paper (https://pubmed.ncbi.nlm.nih.gov/29380929/ ) should be added in discussion.
- In results, information regarding the patients pretreated with steroids should be added- did they have any different pattern of target protein RNA expression?
- In the discussion, authors should state, that the study suffers from low number of patients to fully assess the effects of these markers on survival and needs increased number of patients to assess it to full extent, ideally also at protein level.
Author Response
R: In the introduction, please add information regarding clinical trials studying PD1/PDL1 blockade in human DLBCL and how successful (or not) they were; also add information regarding the PD1/PDL1 antibodies developed to date and clinical trials in dogs
AUTHORS: we added in the introduction data about PD-L1 and PD-1 antibodies in human DLBCL and data from recent in vivo and in vitro studies using anti-PD1 anti PD-L1 antibodies in canine cancers.
R: For data analysis- please include in statistical analysis of also subdivided fractions of PD1 expression (tumour cells and TILs separately). Also add a separate graph showing PD1 expression on tumour cells/TILs. This can actually have a potential clinical relevance if one day PD1/PDL1 therapy will be introduced to veterinary oncology, as PD1 can serve as both tumour suppressor as well as an oncogene, depending on cancer type.
AUTHORS: we agree with the reviewer that a tentative analysis separating fractions of PD-1+ tumor cells from PD-1+ immune cells would be more informative. Therefore, we calculated the tumor proportion score (TPS) and the immune cell density score (IDS) for PD1-1. TPS was calculated as the percentage of PD-1+ tumor cells relative to all viable tumor cells present in the area (positive and negative). The IDS was calculated as the percentage of PD-1+ immune cells relative to all viable tumor cells present in the area (positive and negative). Both TPS and IDS were calculated only in PD-1+ cDLBCLs.
R: The discussion should be expanded to include the potential tumorigenic effects of PD1/PDL1. Additional information, on effect of PDL1 expression on resistance to chemotherapy should also be included and information from this paper (https://pubmed.ncbi.nlm.nih.gov/29380929/ ) should be added in discussion.
AUTHORS: we included a more detailed description of the tumorigenic effects of PD-1 and PD-L1 and results from the paper you suggested.
R: In results, information regarding the patients pretreated with steroids should be added- did they have any different pattern of target protein RNA expression?
AUTHORS: pre-treatment with corticosteroids had no influence on RNAscope scores for any target. This has been added in the text.
R: In the discussion, authors should state, that the study suffers from low number of patients to fully assess the effects of these markers on survival and needs increased number of patients to assess it to full extent, ideally also at protein level.
AUTHORS: this limit has been addressed
Reviewer 2 Report
First of all, I want to thank the possibility of reviewing this interesting article. Its main objective is to analyze the prognostic value of PD-L1, PD-1 and CD8A using alternative methods (RNAscope) due to the scarcity of available antibodies to detect these immune checkpoint proteins in diffuse canine large B-cell lymphomas. To carry out this evaluation, the authors compare the obtained data with clinicopathological characteristics of this type of neoplasm in dog.
Broad comments:
Strengths:
- Use of a less expensive method than others, which can be applied in FFPE tissue samples and allows the integration of molecular information with histopathological characteristics.
- The results suggest that the expression of PD-1 in this type of neoplasm contributes to tumor development that escapes the adaptive immune response. For this reason, it could be used as a therapeutic target. Furthermore, the higher expression of PD-L1 mRNA in dogs with shorter survival and disease-free migration could be used to exclude them from expensive treatments.
Limitations:
- Number of cases analyzed. It would have been interesting to analyze if there was any difference in the expression of these molecules depending on the stage of the disease.
- Recognizing that it is a group that works on this issue, I consider that there are numerous self-citations (10 of 36).
Specific comments
All acronym/abbreviations should be defined the first time they appear in the abstract.
Author Response
Limitations:
- Number of cases analyzed. It would have been interesting to analyze if there was any difference in the expression of these molecules depending on the stage of the disease.
AUTHORS: the limitations of the study have been reported in the discussion, including the small number of cases. About a possible correlation with the stage, this has been done, but unfortunately no significant data were obtained.
Specific comments
-All acronym/abbreviations should be defined the first time they appear in the abstract.
AUTHORS: done
Reviewer 3 Report
This manuscript by Aresu et al. investigates the prognostic value of mRNA expression levels of PD-L1, PD-1, and CD8 in canine diffuse large B cell lymphoma (DLBCL). The authors used RNAscope to measure RNA levels of these potential prognostic markers in canine DLBCL samples and correlated these with several clinically relevant parameters. Several significant associations were found between expression of PD-1/PD-L1 mRNA and disease outcome. These correlations will be useful in the clinic to make decisions about treatment, and this work validates RNAscope as a method for assessing these biomarkers. Major Issues: Several changes could be made to improve the clarity of this manuscript. There is inconsistent use of treatment terms. In the abstract, the treatment groups are referred to as “CHOP” or “CHOP plus APAVAC”. APAVAC is not mentioned again but is replaced with the term “chemoimmunotherapy”. In the second paragraph of the introduction, it is unclear what “two separate profiles”, “the signatures”, and “the clusters” are referring to. Are they all referencing RNA and methylation sequencing? Additionally, although a previous paper is referenced, it may improve clarity to briefly describe “hot” and “cold” in more detail in the materials and methods. The manuscript lacks clarity when discussing comparisons between groups. It is not always obvious what comparison is being made when providing p-values. For example, in the comparison of PD-1 and PD-L1 expression to RNA-seq immune signature two p-values are given. Is this for hot and cold RNA-seq signatures, respectively? Figure 2 M and N are referenced in the first sentence of the results section, but this sentence does not talk about the contents of Figure 2 and N. Additional labels in the panels of Figure 2 may make it easier for readers to interpret the figure (e.g., including the probe and score above or within each image). Also in this section, German shepherd and Rottweiler are specifically mentioned breeds. Are these breeds more likely to develop DLBCL? If not, it is unclear why this detail is relevant. A table with the information of the animals used in the study (paragraph 1 of the results section) may be easier to read, and the most relevant details could be discussed in the text. In the discussion, a stronger emphasis should be placed on the lack of cross-reactive antibodies to highlight the need for alternative modes of detection for these prognostic markers, such as RNAscope. This would underscore the importance of this work and its contribution to the field. Some results merit additional discussion. First, the authors should discuss why TIL group was significantly associated with CD8A expression but not FC CD8+ infiltration as it calls into question which method is more apt for detection of CD8+ T cells within the tumor microenvironment. Second, it is reported that dogs with a hot immune signature had a shorter TTP. Would you not expect that infiltration of lymphocytes would extend TTP? Typically, a hot immune signature is indicative of a host immune response against the tumor. Minor Issues: Figure 3 font size should be increased. Figure 4 – small font size is unclear and light line colors are hard to see. Line 254 – “In the future…” Line 285 – “areas” instead of “area” Line 291 – “to identify” instead of “identifying”
Author Response
R: In the abstract, the treatment groups are referred to as “CHOP” or “CHOP plus APAVAC”. APAVAC is not mentioned again but is replaced with the term “chemoimmunotherapy”.
AUTHORS: throughout the whole manuscript we used the term “chemoimmunotherapy with APAVAC”.
R: In the second paragraph of the introduction, it is unclear what “two separate profiles”, “the signatures”, and “the clusters” are referring to. Are they all referencing RNA and methylation sequencing? Additionally, although a previous paper is referenced, it may improve clarity to briefly describe “hot” and “cold” in more detail in the materials and methods.
AUTHORS: In the introduction we modified the description based on your comment. Also, in the materials and methods we described in details the analysis to separate the two subgroups (“hot” and “cold”) by RNA-seq.
R: The manuscript lacks clarity when discussing comparisons between groups. It is not always obvious what comparison is being made when providing p-values. For example, in the comparison of PD-1 and PD-L1 expression to RNA-seq immune signature two p-values are given. Is this for hot and cold RNA-seq signatures, respectively?
AUTHORS: the two p-values were referred to PD-L1 and PD-1, respectively. This has been better specified in the text
R: Figure 2 M and N are referenced in the first sentence of the results section, but this sentence does not talk about the contents of Figure 2 and N. Additional labels in the panels of Figure 2 may make it easier for readers to interpret the figure (e.g., including the probe and score above or within each image).
AUTHORS: We modified figure 2 following your comments.
R: Also in this section, German shepherd and Rottweiler are specifically mentioned breeds. Are these breeds more likely to develop DLBCL? If not, it is unclear why this detail is relevant.
AUTHORS: Yes, German shepherds and Rottweilers are predisposed to B-cell lymphoma (doi: 10.1016/j.vetimm.2014.02.016 and doi: 10.14202/vetworld.2021.49-55).
R: A table with the information of the animals used in the study (paragraph 1 of the results section) may be easier to read, and the most relevant details could be discussed in the text.
AUTHORS: we included a supplementary table with all clinico-pathological data.
R: In the discussion, a stronger emphasis should be placed on the lack of cross-reactive antibodies to highlight the need for alternative modes of detection for these prognostic markers, such as RNAscope. This would underscore the importance of this work and its contribution to the field.
AUTHORS: we included a brief description of limitations of IHC compared to RNAscope in the discussion
R: Some results merit additional discussion. First, the authors should discuss why TIL group was significantly associated with CD8A expression but not FC CD8+ infiltration as it calls into question which method is more apt for detection of CD8+ T cells within the tumor microenvironment. Second, it is reported that dogs with a hot immune signature had a shorter TTP. Would you not expect that infiltration of lymphocytes would extend TTP? Typically, a hot immune signature is indicative of a host immune response against the tumor.
AUTHORS: we included a brief comment on the absence of correlation between TIL groups and FC CD8+ infiltration.
R: About your second query, very tough question to answer. Basically, the hot signature in tumors is considered the optimum for any therapy triggering the immune system (vaccine, mABs or CAR-T). Unfortunately, splitting our data for the treatment and then prevalence of the signatures, the cases dropped to a very small number and no further statistical analysis/ hypothesis could be performed.
AUTHORS: We are currently working on a larger dataset including whole exome, and hopefully we will have more powerful data shortly, but we prefer not to include comments about this in the paper.
Minor Issues:
Figure 3 font size should be increased. Done
Figure 4 – small font size is unclear and light line colors are hard to see. We modified figure 3 and 4 following your comments.
Line 254 – “In the future...” Done
Line 285 – “areas” instead of “area”. Done
Line 291 – “to identify” instead of “identifying”. Done
Round 2
Reviewer 1 Report
All the suggestions were included in the revised manuscript.